# A Pilot Survey: Oral Function as One of the Risk Factors for Physical Frailty

**DOI:** 10.3390/ijerph19106136

**Published:** 2022-05-18

**Authors:** Ayuto Kodama, Yu Kume, Masahiro Iwakura, Katsuya Iijima, Hidetaka Ota

**Affiliations:** 1Advanced Research Center for Geriatric and Gerontology, Akita University, Akita 010-8543, Japan; ay-kodama@med.akita-u.ac.jp; 2Department of Occupational Therapy, Graduate School of Medicine, Akita University, Akita 010-8543, Japan; kume.yuu@hs.akita-u.ac.jp; 3Department of Rehabilitation Medicine, Akita City Hospital, Akita 010-0933, Japan; masa.iwaku.91@gmail.com; 4Institute of Gerontology, Tokyo University, Tokyo 113-8656, Japan; k-iijima@g.ecc.u-tokyo.ac.jp

**Keywords:** risk of frailty, oral frailty, social support

## Abstract

Background: The aim of this study was to examine the association of the multiple facets of oral, motor, and social functions in community-dwelling older adults, to identify factors that might influence the risk of frailty. Methods: Of the 82 participants included in the study, 39 (5 males and 34 females) were young-old adults, with an average age of 70.5 ± 2.8 years, and 43 (14 males and 29 females) were old-old adults, with an average age of 78.7 ± 2.9 years. We assessed the risk factors for frailty among oral, motor, and social functions. Results: Statistical analysis showed a significant difference in the oral diadochokinesis between the groups (*p* = 0.006). According to the Spearman correlation analysis, a significant association was observed with age and oral diadochokinesis (rs = −0.262, *p* = 0.018), and social support (rs = −0.219, *p* = 0.049). Moreover, binomial logistic regression analysis revealed a significant association of frailty with the occlusal force (odds ratio, 0.031; 95% confidence interval (95% CI), 0.002–0.430; *p* = 0.010), General Oral Health Index (odds ratio, 0.930; 95% CI, 0.867–0.999, *p* = 0.046), and availability of social support (odds ratio, 0.803, 95% CI, 0.690–0.934, *p* = 0.004). Conclusions: To prevent frailty at an early stage, assessments of oral functions, and also that of the availability of social support, are important.

## 1. Introduction

The aging of society is occurring at a more rapid pace in Japan than in any other country. The percentage of the population aged 65 years or over was 23% in 2009, the highest in the world. Moreover, by 2030, one in three people will be 65 years old or over, and one in five will be 75 years old or over [1]. Deterioration in physical health with aging increases the risk of developing disabilities in older adults. The concept of physical frailty proposed by Fried is well-known, and generally refers to slowness, weakness, fatigue, decreased activity, and weight loss. Several previous studies have reported that physical frailty is associated with chronic disease, functional disability, and an increased risk of death [2]. Shimada et al. reported a prevalence of frailty of 11.3% and of pre-frailty of 56.9% among community-dwelling older adults in Japan [3]. On the other hand, it is important to detect frailty in its early stages because it could reversible and a healthy state could be restored with appropriate intervention. Several previous studies have shown that advanced age, cognitive impairment, obesity, presence of multiple diseases, lower educational level, and hospitalization are risk factors for worsening frailty, whereas increased physical activity level, female gender, maintenance of appropriate body weight, low alcohol consumption, high education level, and low baseline disability are associated with a greater likelihood of improvement of frailty [4,5,6,7]. In recent years, while living through COVID-19 countermeasures, many people experienced drastic changes in their ordinary lifestyles, including fearfulness of contracting COVID-19 infection, depression, and excessive sleep [8,9]. The drastic changes associated with the implementation of COVID-19 countermeasures have been implicated in the high incidence of “Corona-Frailty,” and recent longitudinal studies have reported a high proportion of frailty transition in older individuals during the COVID-19 pandemic [10,11]. Furthermore, with a rapidly aging society in Japan, social frailty is a particularly serious issue in community-dwelling older adults in this era of the COVID-19 pandemic. Our previous study [12] reported that the transition rate from robust to social frailty (10.7%) during the COVID-19 pandemic appeared to be higher than that reported by Makizako (8.4%), which was based on a study conducted prior to the onset of the pandemic [13]. This was considered to be attributable to the precautionary measures taken during the COVID-19 epidemic, when community organizations were closed, the elderly were restricted from visiting friends, and social participation was limited [14]. Moreover, a previous study also reported an association between oral functions and frailty [15]. We thought that there is a need to quickly assess older adults at a high risk of frailty from multiple perspectives in the context of the COVID-19 pandemic. Therefore, the purpose of this study was to examine the association of multiple facets of oral, motor, and social functions in community-dwelling older adults during the COVID-19 pandemic, to identify factors that influence the risk of frailty.

## 2. Materials and Methods

### 2.1. Participants

Participants were recruited from Akita city in Akita prefecture, Japan. The target population consisted of 82 persons aged 65 years or older, who were able to walk independently and live at home without assistance for their activities of daily living. The exclusion criteria were: dementia, severe hearing or visual impairment, and intellectual disability. Of the participants, we categorized those aged 65–74 years (5 males and 34 females) into the young-old adult group, and those aged 75 years or older (14 males and 29 females) into the old-old adult group.

Demographic data on age, gender, and height were collected. The study was conducted from 19 November 2021, to 12 January 2022. The study was conducted with the approval of the ethics committee (approval No. 1649).

### 2.2. Outcome

#### 2.2.1. Assessment of Risk of FRAILTY

We assessed the patients for frailty using the Eleven Check questionnaire, developed by the Institute of Gerontology, the University of Tokyo [16,17] (Appendix A). The Eleven Check questionnaire consists of the following 11 items: eating habits (2 items), oral functions (2 items), motor functions (3 items), and social and mental functions (4 items), all of which are important for the maintenance of good health. The response was “yes” or “no,” with a score of 1 for favorable habits and 0 for unfavorable habits, and the total score was 11 (a higher value indicated a lower risk of frailty). Furthermore, persons with a score of 6 or higher were included in the “No risk of frailty” group, and those with a score of 0 to 5 were included in the “high risk of frailty” group.

#### 2.2.2. Assessment of Oral Function

The oral functions were evaluated in terms of the occlusal force, oral diadochokinesis (ODK), and quality of life (QOL) related to oral functions. The occlusal force was checked by palpation of the masseter muscle for contraction by the participants themselves. The ODK was evaluated using the “Kenkou-kun Handy” automatic measuring device (Takei Scientific Instruments Co., Ltd., Niigata, Japan) (Figure 1). The number of repetitions of the monosyllable “ta” per second was recorded. Quality of life related to oral functions was assessed by a self-administered questionnaire called the General Oral Health Index (GOHAI) [18]. Each question had five response categories, with the following scores assigned to each response category (l = always, 2 = often, 3 = sometimes, 4 = rarely, and 5 = never). Scores from positively worded questions were reversed during data processing so that all responses had the same directionality. The GOHAI score was the sum of the responses to the 12 questions, with a high score (maximum = 60) indicating “good oral health”, defined as a good state of internal health.

#### 2.2.3. Assessment of Motor Function

We examined the motor function by the one-leg stand test (OLS), and measurement of the lower leg circumference (LLC), grip strength (GS), and Skeletal Muscle Mass Index (SMI). The subject performed the OLS test by sitting on a chair, lifting the non-dominant leg off the ground, and standing up without recoil, and the examiner determined whether the patient could hold the one-leg stand posture for at least 3 s without recoil. The LLC was measured by placing a measuring tape on the thickest part of the non-beneficial lower leg while the subject sat on a chair. GS was measured using a Smedley-type hand-held dynamometer (Takei Corporation, Niigata, Japan). SMI was calculated by determining the amount of skeletal muscle mass in the limbs using a body composition analyzer (Inner Scan Dual RD-800, TANITA corporation, Tokyo, Japan) and dividing it by the square of height (m).

#### 2.2.4. Assessment of Social Functions

We assessed the social functioning of the subjects in terms of the status of their connection with others, organizational participation, and social support. The status of connection with others was assessed by a self-administered questionnaire based on the Japanese version of the Lubben Social Network Scale shortened version (LSNS-6) [19]. The Japanese version of the LSNS-6 is a social network scale for older adults, consisting of three items related to family networks and three items related to non-family networks. The total score is an equally weighted sum of the scores for the six items, with scores ranging from 0 to 30 (higher scores indicate a more favorable social network, and scores of less than 12 suggest social isolation). Organizational participation was based on the number of ‘yes’ responses to a total of seven items, determined using a two-case method of ‘yes’ and ‘no’ for the existence of an organization to which the respondent belonged, such as a senior citizens’ club or circle: (i) senior citizens’ associations; (ii) organization of health/sports (other than senior citizens’ associations); (iii) organizations of learning/education (other than senior citizens’ associations); (iv) organizations of hobby (other than senior citizens’ associations); (v) neighborhood council; (vi) volunteer organizations; and (vii) income-generating employment or work. The social support situation was assessed using a two-part ‘yes’ or ‘no’ method to determine how many people around supported the subject and how many the subject supported, with the total number of ‘yes’ responses to the total of four items determined as the total score. The questions were as follows: (i) Do you have someone who would listen to your concerns and complaints? (ii) Do you have someone who would help you with housework, shopping, and care/nursing? (iii) Do you listen to others’ concerns or complaints? (iv) Do you help anyone with housework, shopping, and care/nursing?

### 2.3. Statistical Analysis

The Mann–Whitney test was used to compare the age, oral functions (ODK, GOHAI), motor functions (OLS, LLC, GS, and SMI), and social functions (LSNS-6, organizational participation, and social support) between the young-old adults and old-old adults. The gender, occlusal force, and score on the Eleven Check questionnaire were compared using a χ^2^ test. Further, Spearman’s correlation analysis was applied to analyze the relationship between age and oral functions, motor functions, and social functions.

Finally, to determine the factors associated with the risk of frailty, we applied the binomial logistic regression analysis as a dependent variable of the no-risk frailty group, and frailty risk group. The following were used as independent variables for the regression modeling: occlusal force, ODK, GOHAI, OLS, LLC, GS, SMI, LSNS-6, organizational participation, and social support. SPSS Version 27.0 for Windows (SPSS Inc., Chicago, IL, USA) was used for the analysis, and the level of significance was set at *p* = 0.05.

## 3. Results

Of the 82 subjects, there was a significantly higher proportion of women in the young-old adults than in the old-old adults (*p* = 0.034). Table 1 shows a comparison of the oral functions, motor functions, and social functions in young-old adults and old-old adults. According to the Mann–Whitney test, there was a significant difference in the ODK between the groups (*p* = 0.006). On the other hand, while there were no differences in the oral functions, a trend toward more favorable results was observed in the young-old adults as compared to the old-old adults in all categories. There were no differences in the social functions between the groups. Table 2 shows the associations between age and oral, motor, and social functions.

The results of a Spearman’s correlation analysis revealed a significant association for age and ODK (rs = −0.262, *p* = 0.018), and for social support (rs = −0.219, *p* = 0.049) (Table 2, Figure 2 and Figure 3).

Finally, the results of the binomial logistic regression analysis revealed a significant association of frailty with the occlusal force (odds ratio, 0.031; 95% confidence interval (95% CI), 0.002–0.430; *p* = 0.010), GOHAI (odds ratio, 0.930; 95% CI, 0.867–0.999, *p* = 0.046), and social support (odds ratio, 0.803, 95% CI, 0.690–0.934, *p* = 0.004) (Table 3). The goodness of fit of the regression model was also good according to the results of the model χ^2^ test (*p* < 0.01) and Hosmer–Lemeshow test (*p* = 0.593), with a high percentage of correct classifications of 88.3%.

## 4. Discussion

Our study population included a significantly smaller number of young-old adult males than old-old adult males. Previous studies have similarly reported a lower likelihood of social participation among older adult males [20]. Sandra et al. also reported that networks provided by social interactions reduce the risk of physical decline [21]. Thus, our future challenge is to increase the social participation of older male adults. A comparison of the oral, motor, and social functions between young-old adults and old-old adults, revealed significant differences in the ODK between the groups. Hatanaka et al. reported that the prevalence of oral hypofunction in patients aged 75 years or older was over 75%, and in those patients aged 85 years or older, it was over 90%; thus, ODK was significantly negatively associated with age [22]. ODK less than 6.0 times/s is considered to represent impaired oral function according to the definition of oral hypo function proposed by the Japan Geriatric Dental Society. Moreover, some studies have reported a cutoff value of 4.0 times/s or 5.0 times/s to investigate the association between ODK and frailty [23,24]. In this study, the cutoff values were exceeded in both groups; however, the results lend support to those of previous studies showing that ODK declines with age.

Next, a correlation analysis was conducted on the relationship between age and oral and motor, and social functions. The results showed that ODK and social support decreased significantly with age. In the present study, an assessment of the oral functions revealed that only ODK was significantly associated with aging, whereas previous studies suggest that the oral functions, overall, decline with age [22]. Assessments of the occlusal forces and GOHAI in this study are subjective assessments, and it must be borne in mind that the affected individual himself/herself may not be aware of the decline; therefore, early awareness-raising is necessary. Moreover, reduced social support has been reported not only to be associated with aging, but also with the risk of depression and death [25,26]. As the study population in this study mostly consisted of healthy older adults with independent activities of daily living, there may not have been many subjects who needed social support. A previous study has reported that ensuring social support promotes psychological stability among older adults [27]; therefore, it is important to ensure that support is always available.

Finally, to determine the factors associated with the risk of frailty, we conducted binomial logistic regression analysis as a dependent variable of no-risk frailty group, and frailty risk group, and identified a significant association of frailty with occlusal force, GOHAI, and social support. Recent studies have shown a link between oral functions and physical frailty [28,29]. While poor oral functions are a risk factor for a poor health condition, in light of these associations, we propose the concept of oral frailty as an accumulated poor oral status, and its relation to frailty has been studied [30,31]. Hihara et al. defined oral frailty as the fulfillment of three or more of the following conditions: (1) fewer than 20 natural teeth; (2) decreased ability to chew; (3) decreased articulatory oral motor skills; (4) decreased tongue pressure; and (5) substantial subjective difficulties in eating and swallowing [31]. Although physical frailty was not measured accurately in this study, the fact that reduced oral functions were identified as a risk factor for physical frailty suggests that oral frailty might precede physical frailty. Yoshii et al. reported that older adults receiving social support needed help from others due to some physical ailments, and showed physical functional decline [32]. Others have reported that providing social support improves health outcomes [33]. Therefore, in order to reduce the risk of frailty, it is important to ensure that opportunities to provide support as well as receive social support are available. A limitation of this study was while a previous study reported that the occlusal force and ODK rate declined earlier in women than in men [28], there were significantly fewer male participants in this study; it is important to take into account that not only age, but also the gender, influences the risk of frailty. For the sample of this study, a Mann–Whitney test with the number of groups = 2, α = 0.05, power = 80%, and effect size = 0.5 was also insufficient to estimate the sample size of the participants (n = 102) to detect a clinically significant effect [34]. The Japanese government publicly requests community-dwellers to avoid the act of gathering together in a mass or to thoroughly implement social distancing during the ongoing COVID-19 pandemic. The social situations had an undesirable influence on obtaining the epidemiological data on the oral status of more target people in communities. Moreover, while a previous study reported an association between poor oral health (caries, periodontal disease, dry mouth, number of teeth, etc.) and the risk of frailty [15], we did not assess oral health in this study. This is attributed to the fact that dentists were unable to cooperate with the evaluation of oral status under the medical care stringency caused by the COVID-19 pandemic. In addition, the socioeconomic factors that have also been reported to affect physical frailty were not examined in detail (income, education, family/marital status, etc.) [35,36,37]. These requirements should be adjusted for further additional research in the future.

## 5. Conclusions

In this study, we have confirmed that poor oral functions lead to the dilution of social support, and this represents a risk factor for frailty, creating a vicious cycle. To prevent frailty at an early age, regular check-ups and assessments of oral functions, and linking people to healthcare facilities, are necessary.

## Figures and Tables

**Figure 1 ijerph-19-06136-f001:**
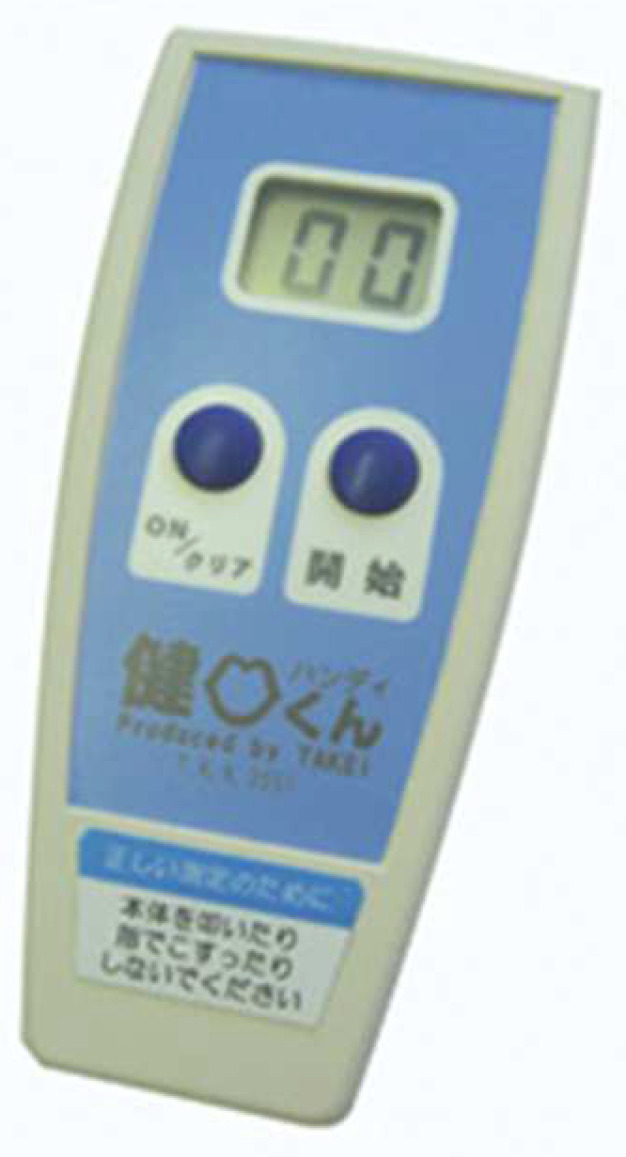
Kenkou-kun Handy (Takei Scientific Instruments Co., Ltd., Niigata, Japan).

**Figure 2 ijerph-19-06136-f002:**
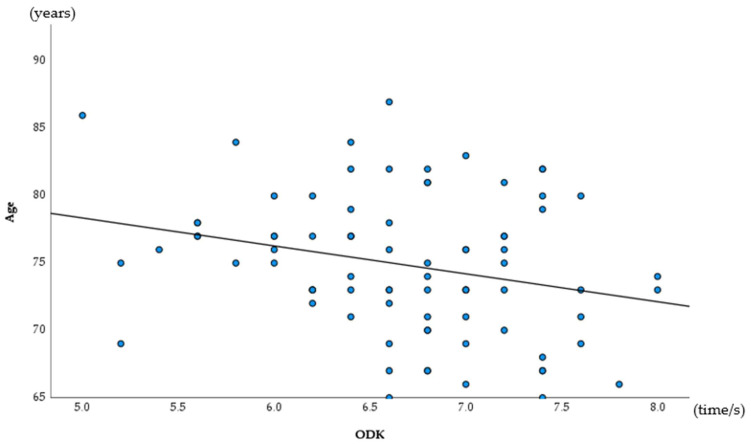
Association for age of ODK (Spearman correlation analysis).

**Figure 3 ijerph-19-06136-f003:**
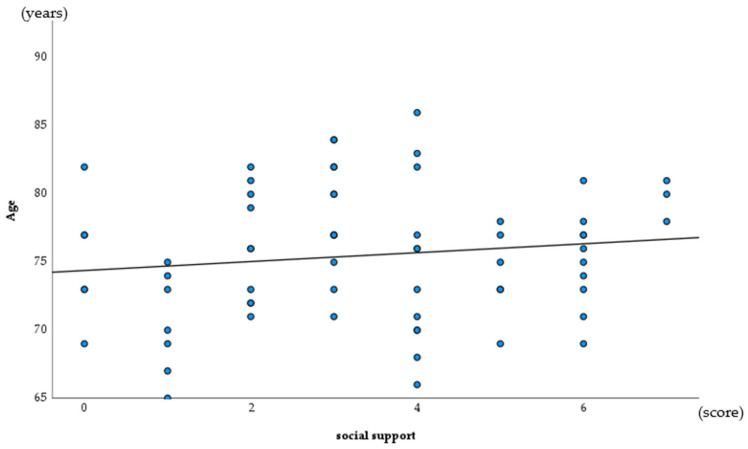
Association for age and social support (Spearman correlation analysis).

**Table 1 ijerph-19-06136-t001:** Comparison of oral, motor, and social functions in young-old and old-old adults.

	Young-Old Adults (n = 39)		Old-Old Adults (n = 43)		*p* Value
	Mean	SD	Mean	SD	
age (years)	70.5	2.8	78.7	2.9	
gender (% female)	87.2		67.4		0.034 *
Eleven Check (% risk)	12.8		14.0		0.880
	Median	IQR	Median	IQR	
**oral functions**					
occlusal force (%)	89.7		95.3		0.330
ODK (times/s)	6.8	0.8	6.5	1.2	0.006 **
GOHAI (point)	57	9	56	6	0.374
**motor functions**					
OLS (%)	56.4		51.2		0.895
LLC (cm)	35.1	3.3	34.5	3.4	0.562
GS (kg)	25.6	5.5	22.5	11.4	0.130
SMI (kg/m^2^)	6.73	0.88	6.55	0.88	0.590
**social functions**					
LSNS-6 (score)	19.5	7	20.5	7	0.161
organizational participation 9 (score)	3	3	3.5	2	0.167
social support (score)	4	1	4	1	0.374

* *p* < 0.05, ** *p* < 0.001, the Mann–Whitney test (age, ODK, GOHAI, OLS, LLC, GS, SMI, LSN-6, organizational participation, and social support), the χ^2^ test (gender, occlusal force, and Eleven Check).

**Table 2 ijerph-19-06136-t002:** Association between age and oral, motor, and social functions.

Eleven Check	Occlusal Force	ODK	GOHAI	OLS	LLC
0.077	−0.06	−0.262 *	−0.058	0.140	−0.005
GS	SMI	LSNS-6	organizational participation	social support	
−0.187	−0.063	0.116	0.100	−0.220 *	

* *p* < 0.05, the Spearman correlation analysis.

**Table 3 ijerph-19-06136-t003:** Models according to binomial logistic regression analysis.

	Coefficient (β)	Odds Ratio	95% CI	*p* Value
occlusal force	−3.485	0.031	0.002, 0.430	0.01 *
GOHAI	−0.072	0.930	0.867, 0.999	0.046 *
social support	−0.220	0.803	0.690, 0.934	0.004 **
constant	8.134			0.004

* *p* < 0.05, ** *p* < 0.01, model χ^2^ test *p* < 0.01, Hosmer–Lemeshow test *p* = 0.593, percentage of correct classifications 88.3%.

## Data Availability

Data available on request due to restrictions, e.g., privacy or ethical. The data presented in this study are available on request from the corresponding author.

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
