# Peer review of "A Pilot Survey: Oral Function as One of the Risk Factors for Physical Frailty"

_ijerph, 2022, doi:10.3390/ijerph19106136_

Round 1

Reviewer 1 Report

It is an interesting manuscript about oral function as one of the risk factors for physical frailty in the elderly.
The ageing of the population does not only occur in Japan, but it is a fact that Japan has one of the world's oldest populations.
Therefore the question arises of how the authors chose the sample size.
If Japan has 23% of the population older than 65, it is strange how the 82 people could be an adequate sample size for this study.
Also, in the section Participants, authors need to add including and excluding criteria.
What does "live at home without assistance" mean? What kind of assistance? Exclusively professional help because assistance could be even living with family members.
The authors use the terms `Young -old adults and old -old adults'. Where are very old? Please refer. Or explain based on what the authors made such a population division.
In addition, in the section on oral function, why authors did not include an oral status?
Dental prostheses?
Also, the introduction is insufficient and does not introduce the reader to the examination issue.

Therefore, although the paper is interesting, it has many deficiencies as the research should be repeated on a larger sample and include the oral status of the respondents.

Author Response

Response to Reviewer 1 Comments

Point 1: If Japan has 23% of the population older than 65, it is strange how the 82 people could be an adequate sample size for this study.

Response 1: Thank you for your supportive comments. According to G power, the sample size required for this study was 102 participants (a Mann-Whitney test with number of groups = 2, α = 0.05, power = 80%, and effect size = 0.5). We have added to the limitation of this study on this issue.

(For the sample of this study, a Mann-Whitney test with number of groups = 2, α = 0.05, power = 80%, and effect size = 0.5 was also insufficient to estimate the sample size of the participants (n = 102) to detect a clinically significant effect [35]. Page8, Line 247)

Point 2:Also, in the section Participants, authors need to add including and excluding criteria.

Response 2: Thank you for supportive comments. As the reviewer suggested, we have added more information excluding criteria.

(The exclusion criteria were: dementia, severe hearing or visual impairment, and intellectual disability. Page 2, Line72)

Point 3: What does "live at home without assistance" mean? What kind of assistance? Exclusively professional help because assistance could be even living with family members.

Response 3: Thank you for supportive reviews comments. "live at home without assistance" mean for activities of daily living. We have added more information.

(The target population consisted of 82 persons aged 65 years or older, who were able to walk independently and live at home without assistance for their activities of daily living. Page 2, Line 72)

Point 4: The authors use the terms `Young -old adults and old -old adults'. Where are very old? Please refer. Or explain based on what the authors made such a population division.

Response 4: Thank you for your supportive comments. We used previous studies to classify the population.

(Of the participants, we categorized those aged 65-74 years (5 males and 34 females) into the young-old adult group, and those aged 75 years or older (14 males and 29 females) in-to the old-old adult group. Page 2, Line74)

(Elpidio A.M. et al. Adverse Effects of Anticholinergic Drugs on Cognition and Mobility: Cutoff for Impairment in a Cross-Sectional Study in Young-Old and Old-Old Adults. Drug Aging 2020, 37, 301-310.)

We have added about information of population division.

Point 5: In addition, in the section on oral function, why authors did not include an oral status? Dental prostheses?

Response 5: Thank you for supportive comments. We were unable to obtain the cooperation of dentists for this study. However, as the reviewer suggested we think it is necessary for us to understand the oral function as well. We have added to the limitations of this study for this issue.

(Moreover, while a previous study reported an association between poor oral health (caries, periodontal disease, dry mouth, number of teeth, etc.) and the risk of frailty [15], we did not assess oral health in this study. Page 8, Line 250)

Point 6: Also, the introduction is insufficient and does not introduce the reader to the examination issue.

Response 6: Thank you for supportive comments. As the reviewer suggested, we have added recent studies to the introduction with more information on the impact under COVID-19 pandemic.

(In recent years, while living through COVID-19 countermeasures, many people experienced drastic changes in their ordinary lifestyles, including fearfulness of contracting COVID-19 infection, depression, and excessive sleep [8, 9]. The drastic changes associated with the implementation of COVID-19 countermeasures have been implicated in the high incidence of "Corona-Frailty,” and recent longitudinal studies have reported a high proportion of frailty transition in older individuals during the COVID-19 pandemic [10, 11]. Furthermore, with a rapidly aging society in Japan, social frailty is a particularly serious issue in community-dwelling old-er adults in this era of the COVID-19 pandemic. Our previous study [12] reported that the transition rate from robust to social frailty (10.7%) during the COVID-19 pandemic appeared to be higher than that reported by Makizako (8.4 %), which was based on a study conducted prior to the onset of the pandemic [13]. This was considered to be attributable to the precautionary measures taken during the COVID-19 epidemic, when community organizations were closed, the elderly were restricted from visiting friends, and social participation was limited [14]. Moreover, a previous study also reported an association between oral functions and frailty [15]. We thought that there is a need to quickly assess older adults at a high risk of frailty from multiple perspectives in the context of the COVID-19 pandemic.  Page 2, Line 46)

Point 7: Therefore, although the paper is interesting, it has many deficiencies as the research should be repeated on a larger sample and include the oral status of the respondents.

Response 7: Thank you for supportive reviews comments. As the reviewer suggested, our study had a small sample size and lacked an assessment of oral status. We would like to conduct additional studies in the future to address those issues. We have added to the limitations of this study for this issue.

(A limitation of this study was while a previous study reported that the occlusal force and ODK rate declined earlier in women than in men [29], there were significantly fewer male participants in this study; it is important to take into account that not only age, but also the gender, influences the risk of frailty. For the sample of this study, a Mann-Whitney test with number of groups = 2, α = 0.05, power = 80%, and effect size = 0.5 was also insufficient to estimate the sample size of the participants (n = 102) to detect a clinically significant effect [35]. Moreover, while a previous study reported an association between poor oral health (caries, periodontal disease, dry mouth, number of teeth, etc.) and the risk of frailty [15], we did not assess oral health in this study. Page 8, Line 244)

Reviewer 2 Report

The manuscript on the subject of oral function and its correlation to physical frailty in elderly population of Akita, Japan has been submitted by the authors for a review and after carefully analyzing it, I have compiled a list of aspects of this work that need to be addressed before possible publication.

My biggest issue with the presented article is, unfortunately, its main subject and aim of this work. In my opinion, it brings very little value and new information to the already existing research and the objective is not sublime enough. The conclusions that were introduced in the end, can be easily drawn even without carrying out the study and seeing its results.

Furthermore, the population chosen for the study could be significantly larger and the inclusion/exclusion criteria are not clearly highlighted. I also think that dividing the patient groups into “young-old” and “old-old” adults is not the right approach in this particular research, especially because the first group is comprised mostly of female participants (~87%) and therefore the sex itself and not just the age could have had an impact on the results. Also, in Lines 54-55 the authors mention that “The demographic data comprised of age, gender and height”. Demographic data in my understanding is socio-economic in its nature and height brings absolutely no merit to the study whilst information such as income, level of education or family/marital status have a huge impact on overall well-being and health of an individual.

The literature presented in the “References” section needs to be updated. There are only 3 papers from 2021 and none from 2022. More articles from the most recent years need to be included, especially due to the impact of COVID-19 pandemic on most people’s socio-economic status since 2020. People were forced to distance themselves from each other, especially the elderly, who were the most endangered group in this situation. This fact has a tremendous influence on the result of this study and needs to be addressed.        

The use of the English language is generally correct and understandable but it needs more attention and editorial work. There are also some minor editorial errors in the text of the manuscript itself:

- Line 33 (the size of the paragraph is too big)

- Line 42 (“overweight” is listed as means to “increase the likelihood of frail improvement” – I believe this mistake is a simple overlooking and obviously this should be corrected to “weight loss or losing weight”, I am not sure that “living alone” should be included in this part as well)

- Lines 52-53 (there is no space before the brackets)

- Line 216 (points (2) and (3) are exactly the same – decreased ability to chew)

Author Response

Response to Reviewer ï¼’ Comments

Point 1: My biggest issue with the presented article is, unfortunately, its main subject and aim of this work. In my opinion, it brings very little value and new information to the already existing research and the objective is not sublime enough. The conclusions that were introduced in the end, can be easily drawn even without carrying out the study and seeing its results.

Response 1: Thank you for your supportive comments. The purpose of our study was to conduct a multidimensional assessment of frailty in a short period of time under the COVID-19 pandemic. As the reviewer suggested, COVID-19 pandemic has a significant impact on the occurrence of frailty. So, we have added more details on the Introduction.

(We thought that there is a need to quickly assess older adults at a high risk of frailty from multiple perspectives in the context of the COVID-19 pandemic. Therefore, the purpose of this study was to examine the association of multiple facets of oral, motor, and social functions in community-dwelling older adults during the COVID-19 pandemic, to identify factors that influence the risk of frailty. Page 2, Line 62)

Point 2:Furthermore, the population chosen for the study could be significantly larger and the inclusion/exclusion criteria are not clearly highlighted. I also think that dividing the patient groups into “young-old” and “old-old” adults is not the right approach in this particular research, especially because the first group is comprised mostly of female participants (~87%) and therefore the sex itself and not just the age could have had an impact on the results.

Response 2: Thank you for supportive comments. As the reviewer suggested, we have added more information excluding criteria. Moreover, the proportion of female in this study was considerably higher. We would like to conduct additional studies in the future to address this issue. We have added to the limitations of this study for this issue.

(The exclusion criteria were: dementia, severe hearing or visual impairment, and intellectual disability. Of the participants, we categorized those aged 65-74 years (5 males and 34 females) into the young-old adult group, and those aged 75 years or older (14 males and 29 females) in-to the old-old adult group. Page 2, Line 72)

(A limitation of this study was while a previous study reported that the occlusal force and ODK rate declined earlier in women than in men [29], there were significantly fewer male participants in this study; it is important to take into account that not only age, but also the gender, influences the risk of frailty. Page 8, Line 244)

Point 3: Also, in Lines 54-55 the authors mention that “The demographic data comprised of age, gender and height”. Demographic data in my understanding is socio-economic in its nature and height brings absolutely no merit to the study whilst information such as income, level of education or family/marital status have a huge impact on overall well-being and health of an individual.

Response 3: Thank you for supportive reviews comments. As the reviewer suggested, demographic data (such as income, level of education or family/ marital status etc) were not examined in this study. We have added to the limitations of this study for this issue.

(In addition, the socioeconomic factors that have also been reported to affect physical frail-ty were not examined in detail (income, education, family/marital status, etc.) [36-38]. These requirements should be adjusted for further additional research in the future. Page 8, Line 252)

Point 4: The literature presented in the “References” section needs to be updated. There are only 3 papers from 2021 and none from 2022. More articles from the most recent years need to be included, especially due to the impact of COVID-19 pandemic on most people’s socio-economic status since 2020. People were forced to distance themselves from each other, especially the elderly, who were the most endangered group in this situation. This fact has a tremendous influence on the result of this study and needs to be addressed.

Response 4: Thank you for your supportive comments. As the reviewer suggested, we have added most recent studies for impact of COVID-19 pandemic.

(In recent years, while living through COVID-19 countermeasures, many people experienced drastic changes in their ordinary lifestyles, including fearfulness of contracting COVID-19 infection, depression, and excessive sleep [8, 9]. The drastic changes associated with the implementation of COVID-19 countermeasures have been implicated in the high incidence of "Corona-Frailty,” and recent longitudinal studies have reported a high proportion of frailty transition in older individuals during the COVID-19 pandemic [10, 11]. Furthermore, with a rapidly aging socie-ty in Japan, social frailty is a particularly serious issue in community-dwelling old-er adults in this era of the COVID-19 pandemic. Our previous study [12] reported that the transition rate from robust to social frailty (10.7%) during the COVID-19 pandemic appeared to be higher than that reported by Makizako (8.4 %), which was based on a study conducted prior to the onset of the pandemic [13]. This was considered to be attributable to the precautionary measures taken during the COVID-19 epidemic, when community organizations were closed, the elderly were restricted from visiting friends, and social participation was limited [14]. Moreover, a previous study also reported an association between oral functions and frailty [15]. Page 2, Line 46)

Point 5: The use of the English language is generally correct and understandable but it needs more attention and editorial work. There are also some minor editorial errors in the text of the manuscript itself:

Response 5: Thank you for supportive comments. Our manuscripts had been checked by a native English speaker.

Point 6:  Line 33 (the size of the paragraph is too big)

Response 6: Thank you for supportive comments. As the reviewer suggested, we have revised the size of the paragraph.

Point 7: Line 42 (“overweight” is listed as means to “increase the likelihood of frail improvement” – I believe this mistake is a simple overlooking and obviously this should be corrected to “weight loss or losing weight”, I am not sure that “living alone” should be included in this part as well)

Response 7: Thank you for supportive reviews comments. As the reviewer suggested, we have revised “weight loss or losing weight”, and have removed “living alone”.

(Several previous studies have shown that advanced age, cognitive impairment, obesity, presence of multiple diseases, lower educational level, and hospitalization are risk factors for worsening frailty, whereas increased physical activity level, female gender, maintenance of appropriate body weight, low alcohol consumption, high education level, and low baseline disability are associated with a great-er likelihood of improvement of frailty [4-7]. Page 1, Line 41)

Point8: Lines 52-53 (there is no space before the brackets)

Response8:Thank you for supportive comments. As the reviewer suggested, we have added the space before the brackets.

Point9:Line 216 (points (2) and (3) are exactly the same – decreased ability to chew)

Response9:Thank you for supportive reviews comments. As the reviewer suggested, we have removed point (3).

(Hihara et al. defined oral frailty as fulfillment of three or more of the following conditions: (1) fewer than 20 natural teeth; (2) decreased ability to chew; (3) decreased articulatory oral motor skills; (4) decreased tongue pressure; (5) substantial subjective difficulties in eating and swallowing [32].  Page 8, Line 233)

Reviewer 3 Report

Nice work, well done!

I  would like to see your results represented in graphics that will sound more clear to the reader.

In line 216, you repeated: (2) decreased ability to chew, (3) decreased ability to chew.

Positive points

The article was well written, the statistical analysis was well done, and it presents similar methodology and content as others on this subject.

Negative points

The authors present data just in tables, which makes their results more difficult to identify.

I suggest that they present also graphics, with significance, to become easier for the reader to relate data at the first sight.

Moreover, the Authors suggested that oral conditions can lead to frailty, but they did not perform an intraoral examination.

How many individuals had a toothache?

How many were totally or partially edentulous?

How many had removable prostheses?

These conditions affect the occlusal force, which should have been taken into account and measured by an appropriate device like an occlus-o-guide for example, or an electromyograph, their approach just feeling the contraction of the masseter muscle, as they mentioned, was subjective, and can not be measured, that is not scientific.

How many individuals had periodontal problems, such as gingival pockets, bleeding gums, etc?

Aging patients present a higher risk of developing bacteremia, which can affect not just oral, but general health, more than just heart problems, aging patients presenting infections can evolve into temporary dementia.

Conclusion:

The general overview is that the article is good, well written, and similar to others as I first mentioned, but could be improved to become more free-flowing.

I will not suggest a different methodology for this paper, but intraoral examination and a different methodology for measuring occlusal force should be considered for future papers.

Author Response

Response to Reviewer 3 Comments

Point 1: The authors present data just in tables, which makes their results more difficult to identify. I suggest that they present also graphics, with significance, to become easier for the reader to relate data at the first sight.

Response 1: Thank you for your supportive comments. As the reviewer suggested, we have added figure about for results of the Spearman correlation analysis. (Page 6-7)

Point 2:2.Moreover, the Authors suggested that oral conditions can lead to frailty, but they did not perform an intraoral examination. How many individuals had a toothache? How many were totally or partially edentulous? How many had removable prostheses? These conditions affect the occlusal force, which should have been taken into account and measured by an appropriate device like an occlus-o-guide for example, or an electromyograph, their approach just feeling the contraction of the masseter muscle, as they mentioned, was subjective, and can not be measured, that is not scientific. How many individuals had periodontal problems, such as gingival pockets, bleeding gums, etc?

Response 2: Thank you for supportive comments. As the reviewer suggested, we were unable to obtain the cooperation of dentists for this study. However, as the reviewer suggested we think it is necessary for us to understand the oral function as well. We have added to the limitations of this study for this issue.

(Moreover, while a previous study reported an association between poor oral health (caries, periodontal disease, dry mouth, number of teeth, etc.) and the risk of frailty [15], we did not assess oral health in this study. Page 8, Line 250)

(These requirements should be adjusted for further additional research in the future. Page 8, Line 254)

Point 3: Aging patients present a higher risk of developing bacteremia, which can affect not just oral, but general health, more than just heart problems, aging patients presenting infections can evolve into temporary dementia.

Response 3: Thank you for supportive reviews comments. We did not examined the current medical and past history for participants. We would like to conduct additional studies in the future to add the information of current Medical and Past History.

Point 4: I will not suggest a different methodology for this paper, but intraoral examination and a different methodology for measuring occlusal force should be considered for future papers.

Response 4: Thank you for your supportive comments. As the reviewer suggested, our study had a lacked an assessment of oral status. We would like to conduct additional studies in the future to address this issue. We have added to the limitations of this study for this issue.

(Moreover, while a previous study reported an association between poor oral health (caries, periodontal disease, dry mouth, number of teeth, etc.) and the risk of frailty [15], we did not assess oral health in this study. Page 8, Line 250)

(These requirements should be adjusted for further additional research in the future. Page 8, Line 254)

Round 2

Reviewer 1 Report

The authors corrected the manuscript, although they did not accept all the comments. I can recommend it for publication, but only if it is published as a pilot study due to the insufficient number of respondents.

Author Response

Point: The authors corrected the manuscript, although they did not accept all the comments. I can recommend it for publication, but only if it is published as a pilot study due to the insufficient number of respondents.

Response: Thank you for supportive comments. As the reviewer suggested, we have added words of "A pilot survey" towards the previous title. We also recognizes that increasing the number of participants should be considered in the future investigation, and will keep examining longitudinally in communities.

(A pilot survey: Oral function as one of the risk factors for physical frailty Page 1, Line 2)

Reviewer 2 Report

I am pleased to see that the authors were very considerate of the given feedback and meticulously addressed the issues that I have found with the paper. Given my main concern regarding the manuscript, which is - does it bring new valuable information and fill in the gap in the already existing literature? There were a lot of limitations of the study such as little number of patients participating or insufficient data in some areas, the authors should also include a paragraph that explains those limitations in more detail, especially the fact that conducting research during the ongoing COVID-19 pandemic presented difficulties in obtaining the information that could otherwise be easily accessible. All the reasons for not including data on e.g. the patients’ oral status (the use of dentures, periodontal diseases etc. and other chronic diseases) must be included in the text.

Author Response

Point1: I am pleased to see that the authors were very considerate of the given feedback and meticulously addressed the issues that I have found with the paper. Given my main concern regarding the manuscript, which is - does it bring new valuable information and fill in the gap in the already existing literature? There were a lot of limitations of the study such as little number of patients participating or insufficient data in some areas, the authors should also include a paragraph that explains those limitations in more detail, especially the fact that conducting research during the ongoing COVID-19 pandemic presented difficulties in obtaining the information that could otherwise be easily accessible. All the reasons for not including data on e.g. the patients’ oral status (the use of dentures, periodontal diseases etc. and other chronic diseases) must be included in the text.

Response2: Thank you for supportive comments. As the reviewer suggested, we have added the follow information regarding the difficulty that we could not predict during the COVID-19 pandemic in Japan.

(The Japanese government publicly requests community-dwellers to avoid the act of gathering together in a mass or to thoroughly implement social distancing during the ongoing COVID-19 pandemic. The social situations had undesirable influence on obtaining the epidemiological data of the oral status in more target people in communities. Page 8, Line261)

(This is attributed to the fact that dentists were unable to cooperate with the evaluation of oral status under the medical care stringency caused by the COVID-19 pandemic. Page 8, Line 267)
